# Differences in Formation of Prepuce and Urethral Groove During Penile Development Between Guinea Pigs and Mice Are Controlled by Differential Expression of Shh, Fgf10 and Fgfr2

**DOI:** 10.3390/cells14050348

**Published:** 2025-02-27

**Authors:** Shanshan Wang, Zhengui Zheng

**Affiliations:** Department of Biomedical Sciences, Southern Illinois University School of Medicine, Carbondale, IL 62901, USA; swang24@siumed.edu

**Keywords:** preputial development, urethral groove formation, guinea pig penile development, Sonic hedgehog, Fgf signal

## Abstract

The penile tubular urethra forms by canalization of the urethral plate without forming an obvious urethral groove in mice, while the urethral epithelium forms a fully open urethral groove before urethra closure through the distal-opening-proximal-closing process in humans and guinea pigs. Our knowledge of the mechanism of penile development is mainly based on studies in mice. To reveal how the fully opened urethral groove forms in humans and guinea pigs, we compared the expression patterns and levels of key developmental genes using in situ hybridization and quantitative PCR during glans and preputial development between guinea pigs and mice. Our results revealed that, compared with mouse preputial development, which started before sexual differentiation, preputial development in guinea pigs was delayed and initiated at the same time that sexual differentiation began. *Fgf10* was mainly expressed in the urethral epithelium in developing genital tubercle (GT) of guinea pigs. The relative expression of *Shh*, *Fgf8*, *Fgf10*, *Fgfr2,* and *Hoxd13* was reduced more than 4-fold in the GT of guinea pigs compared to that of mice. Hedgehog and Fgf inhibitors induced urethral groove formation and restrained preputial development in cultured mouse GT, while Shh and Fgf10 proteins induced preputial development in cultured guinea pig GT. Our discovery suggests that the differential expression of *Shh* and *Fgf10/Fgfr2* may be the main reason a fully opened urethral groove forms in guinea pigs, and it may be similar in humans as well.

## 1. Introduction

The mammalian penis develops from the bisexual precursor called the genital tubercle (GT) under the influence of androgens [1,2,3,4]. Compared with mice and rats, humans and guinea pigs form fully opened urethral grooves at the end of the bisexual stage, and the process of tubular urethral formation follows distal-opening-proximal-closing, that is, the “Double Zipper” model [1,3]. In humans, the penile urethra develops via an “Opening Zipper” by canalization of the solid urethral plate to form the open urethral groove before 9.5 weeks of gestation. Then, androgen-induced proximal closure of the urethral groove starts tubular urethral formation in the penile shaft before 10.5 weeks [1]. Human preputial development is initiated at 10–11 weeks of gestation, which is almost the same time when tubular urethra starts to form in the proximal region. Using the guinea pig model, we found that cell proliferation in the outer layers (including the basal layer) and programmed cell death in the inner layers of urethral epithelium play key roles during dorsal-to-ventral displacement and final opening of the urethral canal to form a urethral groove; these processes show no difference between males and females [5]. In mice, the urethral epithelium forms a urethral plate; a urethral opening can be observed in the proximal region at E13.5. From the urethral opening to the distal tip, the ventral part of the urethral epithelium has never formed an open urethral groove during fetal development, and thus there is no distal-opening-proximal-closing process during tubular urethral formation [6]. The preputial swellings appear as secondary outgrowths on the lateral edges of the tubercle at E13.5 and continue to grow laterally and ventrally to form the prepuce [6]. At E15.5, the preputial swelling covers the proximal region of the glans before sexual differentiation [6]. Clearly, human preputial development is initiated at the same time as the androgen-dependent-proximal-closing of urethral groove initiation, while in mice, the preputial development is initiated relatively earlier and begins at the bisexual stage of the external genital development; unfortunately, the mechanism causing the difference is unknown.

Theories on the development of human prepuce fall into two opposing ideas. One idea is that the development of the human prepuce involves the formation and distal extension of the preputial folds to eventually completely cover the glans. This idea is exemplified by Hunter who described “folds of ectodermal tissue that appear to flow over the dorsum of the glans as the beginning of the prepuce then extends distally to cover the glans” [7]. An alternate theory is that a dorsal skin fold forms a preputial fold and extends distally to cover the glans; at the same time, epithelial ingrowth occurs to form the preputial lamina. This hypothesis was proposed by Glenister in 1956 [8]. The two theories were illustrated recently by Cunha et al. [9]. Liu et al. [10] found that the prepuce initially forms on the dorsal aspect of the glans at approximately 12 weeks of gestation. After sequential proximal-to-distal remodeling of the ventral urethral plate along the ventral aspect of the glans, the prepuce of epidermal origin fuses in the ventral midline. Cunha et al. [9] proposed a novel morphogenetic mechanism for the formation of the preputial lamina, namely the splitting of the thick epidermis of the glans into the preputial lamina and the epidermis via the intrusion of mesenchyme, finding that the process begins on the proximal aspect of the glans and extends distally. The cellular and molecular mechanisms of preputial development remain unknown.

The development of external genitalia is controlled by local developmental genes, such as fibroblast growth factors (Fgfs), Sonic hedgehog (Shh), Bone morphogenetic protein (Bmp), Wnt, and several Hox genes [11,12,13]. Several *FGF* genes, including *FGF8*, *FGF10*, and fibroblast growth factor receptor 2 (*FGFR2*), were found to be expressed in the foreskin of children with hypospadias [14]. Mutation in *Fgf10* or *Fgfr2* induces genital malformation in mice [15,16]. It has been shown that *Bmp4* is required for the initiation of GT outgrowth [17]. Additionally, knockout of an upstream regulator of *Bmp4*, *Isl1,* in murine shows abrogated genital outgrowth [18]. Several WNT ligands are expressed in the developing mouse GT; canonical WNT signaling is required in normal murine GT outgrowth [19,20]. Wnt5a is one of the main WNT ligands regulating urethral tube formation as well as external genitalia outgrowth in mice [21]. Human autosomal dominant mutations in *WNT5A* cause Robinow Syndrome, which shows micropenis and hypoplastic scrotum in males, and hypoplasia of the clitoris and labia in females [22]. Extensive studies of *Hoxd13* and *Hoxa13* have demonstrated their essential role in the development of the GT, double mutants exhibit agenesis of the GT, and heterozygosity for *Hoxa13* or *Hoxd13* causes patterning defects of the phallus [23]. Human mutations in the *HOXA13* gene are responsible for the range of phenotypes observed in hand-foot-genital syndrome [24]. Homozygous mutations of *HOXD13* in men exhibit hypospadias [25]. *Shh* is one of the most intensively studied genes in external genital development. It is a major morphogenic regulator of the outgrowth of the GT mainly through the regulation of cell cycle progression [6,26]. A significant decrease in mRNA expression of the *SHH* and *PTCH1* genes was found in boys with proximal hypospadias compared with boys without hypospadias [27]. *Shh* expression has been detected in the urethral plate in the developing GT of guinea pigs. Compared to mice, the *Shh* expression domain on the ventral side of the developing GT in guinea pigs extends out to the ventral surface epithelium [5].

Here, we report the results of experiments aimed at identifying differentially expressed genes in a guinea pig model that more closely resembles human penile development, and we tested the function of the selected genes in preputial development and urethral groove formation using organ culture. Since a detailed interpretation of guinea pig penile development at the stage of distal-opening-proximal-closing of the urethra is not available, we first present a detailed embryological study of genital development between E27 to E33. Preputial development is initiated before sexual differentiation occurs in mice but after sexual differentiation starts in guinea pigs; we compare key genes’ expression patterns and levels between guinea pigs and mice; finally, we demonstrate that *Shh* and *Fgf10* play critical roles in preputial development and urethral groove formation during distal-opening-proximal-closing of penile development.

## 2. Materials and Methods

### 2.1. Animals and Treatments

Sexually mature Hartley guinea pigs were purchased from Elm Hill Labs, and ICR mice were purchased from Envigo RMS Inc. (Indianapolis, IN, USA). Guinea pigs and mice were housed in a pathogen-free barrier facility on 12 h light/dark cycles, with access to food and water ad libitum. All experiments were conducted in accordance with the “Guide for the Care and Use of Laboratory Animals”. The experimental protocols (Guinea pigs: 20-014, mice: 23-011) were approved by the Institutional Animal Care and Use Committee of Southern Illinois University Carbondale. Time-mating of guinea pigs was achieved based on a previously established method [3]. A minimum of 2 litters of embryos was collected at each stage, and GTs were dissected under a stereoscope. The sex of the embryos at E30 and later stages was identified by inspection of gonadal morphology under the stereoscope, while the sex of the embryos at earlier stages was identified by genotyping using Y (*Sry*) and X (*Dystrophin*) chromosome-specific genes [28]. To detect cell proliferation in guinea pigs, 5-bromo-2′-deoxyuridine (BrdU, 50 mg/kg) was injected intraperitoneally, and embryos were collected 4 h later for immunohistochemistry.

### 2.2. Organ Culture and In Vitro Cyclopamine, BGJ-398, Shh, and Fgf-10 Administration

Mouse and guinea pig embryonic GT (at E14.5 and E27, respectively) culture setup followed the previously described method [29]. Methyltestosterone (MT) was dissolved in ethanol (0.05 M stock, filter sterilized) and then diluted into the medium to 10 nM. Cyclopamine (Cat#: A8340), BGJ398 (Cat#: A3014), and mouse Shh protein (Cat# P1230) were purchased from ApexBio (Houston, TX, USA). The Cyclopamine stock solution (10 mM in DMSO) was diluted to the final concentration (200 nM) with culture medium before use. BGJ398 stock solution (5 mg/mL in DMSO) was diluted to 1.4 nM to inhibit Fgf-10/Fgfr2 using a culture medium. Mouse Fgf-10 protein was purchased from R&D systems (Cat# 6224-FG-025/CF, Newark, DE, USA). Shh and Fgf-10 proteins were added into the culture medium with the final concentration of 100 ng/mL and 50 ng/mL, respectively. The GTs were cultured for 48 h and then processed for morphology analysis.

### 2.3. Histology and Immunohistochemistry

Guinea pig and mouse embryos at different stages were harvested and rinsed with PBS. Paraffin sections of GT were prepared and stained with hematoxylin and eosin, as we previously described [3]. Immunohistochemistry was performed using anti-BrdU (G3G4, Cat# AB 2314035, RRID: AB_2618097, DSHB, Iowa City, IA, USA) according to procedures modified from the previously established immunofluorescence protocol [5]. BrdU antibody was detected using ABC Kit (PK-6100, Vector Laboratories, Newark, CA, USA) according to the manufacturer’s operation manual. To quantify BrdU-positive cells in the preputial initiation region of developing E29 guinea pig GT, image stacks were acquired using the 40× objective, and ImageJ.JS software (version 1) was used to identify and count the positive cells in newly formed preputial mesenchyme and surrounding epithelium. Since there were only 3 serial sections (6 μm) of the preputial formation starting area in each GT, positive cells in the interested structure of every section in 3 different GTs were counted. The Fgf-10 immunofluorescence in guinea pig and mouse GTs was performed following the established method in our laboratory [3,5,29] using FGF-10 (H121) polyclonal antibody purchased from Santa Cruz (Cat# SC-7917, RRID: AB_2262731, Dallas, TX, USA). Sample size, *n* = 3 litters, slides obtained from 3 males (one from each litter) were selected for immunostaining.

### 2.4. In Situ Hybridization

Three embryos of different stages were selected for in situ hybridization analysis. In situ hybridization was performed as described [30] with some modifications [5]. To create the RNA probes, the cDNA of developmental genes was PCR-amplified from embryonic guinea pig cDNA using the primers designed with PrimerQuest software (version RUO22-1233-001) from IDT-DNA (https://www.idtdna.com/pages/tools/primerquest, accessed on 6 November 2022), and all primers are listed in Appendix A. These were then cloned into the pGEM**^®^**-T Easy Vector (Cat# A1360, Promega, Madison, WI, USA) according to operation manual, except for *Fgf10*. The guinea pig *Fgf10* IMAGE clone was purchased from GenScript (Clone ID: ODo12835, Piscataway, NJ, USA). Cloned DNAs were then amplified using M13 primers, and a reverse transcription reaction was performed using the DIG RNA Labeling Kit (SP6/T7: #EP0131/#EP0111, Thermo Scientific, Waltham, MA, USA) from Roche (Cat#: 11175025910, Indianapolis, IN, USA) to produce antisense RNA probes. A guinea pig *Fgf10* RNA probe was synthesized using T3 RNA polymerase (#EP0101, Thermo Scientific, Waltham, MA, USA).

### 2.5. Quantitative Gene Expression Analysis Using RT-QPCR

Total RNA was extracted from male GT of E12.5 and E13.5 mice, as well as E23 and E26.5 guinea pigs, using the TRIzol method according to the operation manual (Cat: 10296010, Invitrogen, Carlsbad, CA, USA). RNA quality was assessed following the published method [31]. The cDNA was synthesized from 500 ng total RNA using iScript Reverse Transcription Supermix (Bio-rad, Hercules, CA, USA). Primers for all guinea pig genes were designed using the PrimerQuest Tool (Integrated DNA Technologies, Inc) to amplify cDNAs of around 90–150 bp sequences, and all exhibited similar amplification efficiency (r ≥ 97) as assessed by the amplification of control cDNA dilution series. Primers for mouse genes were designed and validated by OriGene. Primer sequences are summarized in Appendix A. Quantitative PCR was performed using a CFX96 Real-Time PCR Detection System (Bio-rad) with iQ SYBR Green Supermix (Bio-rad) as the detector. The Real-Time QPCR was programmed for 3 min at 95 °C followed by 40 repetitive cycles of melting (94 °C), annealing, and extension (60 °C) for 10 and 20 sec, respectively. The cycle threshold (Ct) values were used to calculate the relative steady-state levels of specific mRNA in the samples. After amplification, the specificity of the PCR was determined by both melt curve analysis and gel electrophoresis to verify that only a single product of the correct size was present. Data were normalized against a housekeeping gene, glyceraldehyde 3-phosphate dehydrogenase (*Gapdh*), using the ∆∆Ct method [32].

### 2.6. Statistical Analysis

Cell proliferation and RT-qPCR data were subjected to statistical analysis using SPSS 22.0 software. Quantitative data were presented as mean ± standard error (mean ± SE). Paired *t*-tests were used for comparisons. Statistical significance in the results was noted: * *p* ≤ 0.05, ** *p* ≤ 0.01, *** *p* ≤ 0.001.

## 3. Results

### 3.1. Guinea Pigs Show Delayed Preputial Formation During the Glans and Preputial Development Compared with Mice

The early development of the glans penis before sexual differentiation is similar between mice and guinea pigs [3,6]. The difference can be observed at a comparable stage of around E13.5 in mice and around E26.5 in guinea pigs. In guinea pigs, the open urethral groove is first observed at E28 [3]. Based on the limb and external genital development, the stage of E27 in guinea pigs matches E13.75 in mice (Figure 1A,I). At this stage, the shape of the urethral plate from distal (reverse triangle) to proximal (triangle) is also similar between guinea pigs and mice, except that guinea pigs have a larger urethral canal (Figure 1B–H,J–O). The development of preputial swellings is initiated at E13.5, and preputial glands can be observed at around E14.5 in mice [6]. The epidermal epithelium folds to form the preputial lamina on either lateral side of proximal GT in E13.75 mice, which subsequently separates the glans and preputial mesenchyme (Figure 1L,M).

In guinea pigs, no preputial swelling was observed, and no preputial lamina or preputial glands were found in all transverse sections from proximal to distal GT at E27 (Figure 1A–H). We observed a mouse preputial swelling-like structure in E29 guinea pigs (Figure 2A), and preputial development is initiated around the same time as the initiation of sexual differentiation (Figure 2A–G and Ref. [3]). The epidermal epithelium folds to form the preputial lamina from both lateral sides in the proximal region and separates the glans from the preputial mesenchyme in E29 GT (Figure 2A–H). Compared to the sections in Figure 2G,H, we can see that epithelial ingrowth exists during the formation of the preputial lamina in guinea pigs. The opened urethral groove emerges at E28, and sexual differentiation is initiated at E28 to E29 in guinea pigs [3]. In mice, sexual differentiation of external genitalia is observed around E16, and the penile masculinization stage of E16.5 mice is comparable to that of E29 guinea pigs (Figure 2A–O). We can clearly see that the closing of the urethral tube is initiated in the proximal region of the E29 GT (Figure 2F–H), and then progressively extends distally, reaching the distal tip of the glans penis by E33 (Figure 3D–K).

As Figure 2 shows, we can see the fully opened urethral groove in the middle region of the glans penis (Figure 2E), not fully opened urethral epithelium with the urethral canal in the more distal region (close to the distal tip) (the opening zipper, Figure 2C,D), and the closed tubular urethra in the proximal region (the closing zipper, Figure 2F,G) in E29 guinea pigs. In E16.5 mice, the tubular urethra has formed in the proximal region (Figure 2O), indicating this is a comparable penile masculinization stage to the E29 guinea pig. In the middle region of the E16.5 mouse penis, the urethral epithelium in the glans region forms a urethral plate without a urethral canal. The medial boundary of epidermal epithelium-derived preputial epithelium from either side, together with the ventral surface epithelium of the GT, forms a shallow groove (Figure 2L,M), and a urethral groove-like structure is found only in the distal region without preputial covering (Figure 2J). The prepuce has covered more than half of the glans and reached the distal region in E16.5 mice (Figure 2K–O), but only a very small portion of the proximal region in E29 guinea pigs (Figure 2G,H). Interestingly, we found that in the mid-distal region of the E33 guinea pig penis, the epidermal epithelium has increased layers, and the thick epidermis is split and remodeled by the intrusion of preputial mesenchyme to form the preputial lamina and a thin surface epidermis (Figure 3F–N). When we compare the proximal (Figure 3I–K) with the distal (Figure 3D–H) region, the thicker epithelium in the distal region is apparently noticeable.

The evaginating preputial mesenchyme can be observed from the mid-distal boundary into the distal (Figure 3H,G,L–N). Similar processes can also be found in E16.5 mice (Figure 2L). In both E16.5 mice and E33 guinea pigs, we can see that the prepuce has fully covered the middle region of the glans penis, and the preputial lamina has almost formed a circular (tubular) shape, separating the preputial and glans mesenchyme in the same region (Figure 2M and Figure 3I,J). The only differences are the frenulum on the ventral side of the E33 guinea pig penis (Figure 3A,I,J), and the still unclosed urethra in the middle glans region of the E16.5 mouse penis (Figure 2M). Only fractional preputial lamina can be observed in the proximal (Figure 2N and Figure 3K) of the glans penis at this stage, which suggests that the epidermal epithelium might invaginate toward the proximal region of the glans penis to form preputial lamina in both guinea pigs and mice.

The preputial glands in guinea pigs develop even later, and no preputial glands have been found at any penile developmental stages before E30 (Figure 3A–H; Ref. [3]). The earliest stage we observed preputial glands in guinea pigs is E33, when the preputial glands are seen in the proximal penis (Figure 3J,K). In addition, the epithelial tag can be distinguished in the distal-most region of E29 (Figure 2A) and E33 (Figure 3A–C) guinea pig penises, but not in the penises of mice at comparable stages.

### 3.2. Differential Cell Proliferation in Epithelium and Mesenchyme Contributes to Preputial Development

In light of our findings that both epithelial and mesenchymal cells may invaginate/evaginate to form the preputial lamina and prepuce during penile development, we hypothesized that differential cell proliferation may exist and contribute to epithelial ingrowth and mesenchymal evagination during preputial lamina formation in guinea pigs. To test this hypothesis, we performed a BrdU labeling cell proliferation assay. Figure 4E is the transverse section of the proximal (initiation of preputial development) GT of E29 guinea pigs. We found that there were more BrdU-positive cells (3.1 times, *p* = 0.0026, *n* = 3) in the epithelium than in the surrounded mesenchyme, and the epithelium invaginated and split the mesenchyme (Figure 4E,F,I). In a slightly distal section (Figure 4C), even more BrdU-positive cells (5.6 times, *p* = 0.0007, *n* = 3) were detected in the epithelium than in the mesenchyme; the epithelium increased in layers, and a small portion of mesenchymal cells was separated from the glans mesenchyme and formed the precursor cells of preputial mesenchyme (Figure 4C,D,H). In a more distal section (Figure 4A), the majority of BrdU-positive cells were restricted in the preputial mesenchyme and the epithelial layer adjacent to the glans mesenchyme (Figure 4A,B,G). The results indicate that epithelial cell proliferation leads to an increase in epithelial layers, and the evagination of preputial mesenchyme into the thickened distal epithelium is due to mesenchymal cell proliferation. In the distal region of the E33 penis, the majority of BrdU-positive cells are located in the basal epithelial layer adjacent to the glans mesenchyme and developing preputial mesenchyme (Figure 4J,K). In the proximal region of the E33 penis, more BrdU-positive cells are located in the basal layer of the preputial lamina epithelium (Figure 4L). The results indicate that the fractional preputial lamina in the proximal region of the E33 penis (Figure 3K) is mainly derived from epithelial invasion.

### 3.3. The Expression Patterns of Key Developmental Genes in Guinea Pig Genital Tubercle

We next sought to identify the molecules that mediate GT development in guinea pigs and performed a comparison with previously identified important genes in mouse GT development. We carried out an in situ hybridization screening on guinea pig GTs of those developmentally important genes identified from mice, focusing on the expression of *Shh*, *Fgf*, *Bmp*, *Wnt,* and *Hox* genes. We first present key gene expression patterns of these pathways in E23-23.5 guinea pig GTs, which are comparable to E12.5 mouse GTs.

Previous work has reported the presence of *Shh* in developing guinea pig GT and found that the expression pattern was slightly different from that of mice at multiple stages [5]. Here, we show that at E23, *Shh* is expressed in the urethral epithelium of guinea pig GT (Figure 5A), and the pattern is similar to that of E12.5 mice [6]. The *Shh* receptor gene *Ptch1* is expressed in a broad domain of the mesenchymal cells surrounding the *Shh*-expressing urethral epithelium at E23 in guinea pigs (Figure 5B), which is also similar to that of E12.5 mice [6]. In mouse GT, *Shh* is expressed in the urethral epithelium from E11.5 to E14.5 and in preputial glands from E13.5 to E14.5 [11,33]. To identify *Shh* expression at later stages in guinea pig GT, we performed *Shh* in situ hybridization from E27 to E33. *Shh* expression in the urethral plate can be detected as late as E28, and no preputial expression of *Shh* has been observed at any stage before E28 (Figure 6A–C and Ref. [5]). The earliest stage at which preputial *Shh* expression can be detected is around E32, when *Shh* is initially expressed in developing preputial glands as small dots in both male and female GT (Figure 6D,E).

*Hoxd13* is expressed in the distal mesenchyme of E23.5 guinea pig GT (Figure 5C,D), and the expression pattern resembles that of E12.5-14.5 mouse GT [6,34]. Interestingly, the expression of *Hoxd13* in GT is relatively weaker compared with the strong expression in the hindlimbs of guinea pigs (Figure 5C,D).

*Bmp4* is expressed strongly in the distal tip mesenchyme of E23.5 guinea pig GT (Figure 5E,F), similar to that of E12.5 mouse GT [6]. The expression on the ventral side is localized to the mesenchyme surrounding the urethral epithelium, resembling the expression of *Ptch1* in this region (compare Figure 5B,E). Additionally, *Bmp4* expression on the dorsal side resembles the expression of *Hoxd13* in the same region (compare Figure 5D,F). *Bmp7* and *Bmp4* have similar expression patterns in E23.5 hindlimbs (Figure 4E, H), but their expression patterns in GT are quite different. *Bmp7* is expressed in both the urethral epithelium and mesenchyme of E23.5 GT (Figure 5G,H). The urethral expression is weak and confined only to the partial urethral epithelium near the distal tip, and the mesenchymal expression is not like that of *Bmp4* and *Ptch1*, which are expressed in cells adjacent to the urethral epithelium but further away from the urethra, mainly on the side of the distal tip (Figure 5G,H).

*Fgf8* expression is detected in a subset of the *Shh* expression domain in the guinea pig GT and is restricted to the anterior region of the urethral epithelium at E22 (Figure 5J). It remains at the distal tip of the urethral plate when the tubercle grows out at E23 and can only be seen from a dorsal view (Figure 5K,L). The pattern of *Fgf8* in guinea pig GT also resembles the expression in mice [6,35]. Similarly to *Hoxd13*, when compared with the strong *Fgf8* expression in limb buds, the signal is much weaker in developing GT.

*Fgf10* is mainly restricted to the urethral plate in E23 guinea pig GT (Figure 5M,N), which differs from the reported mesenchymal expression in mouse GT [11,15]. The expression of the Fgf receptor *Fgfr2* in E23 guinea pig GT is also found in the urethral epithelium (Figure 5O,P). The expression pattern of *Fgfr2* in the guinea pig urethra is similar to that in mice at early stages [16], but there is a lack of preputial expression before E28 compared with mice at similar developmental stages (E13.5–14.5) due to the absence of preputial development until E29. As the urethral plate expression of *Fgf10* is not the same as reported in mouse GT, we performed *Fgf10* and *Fgfr2* in situ hybridization on the GT of E22, E23.5, and E26 guinea pigs (comparable to E11.5, E12.5, and E13.5 mice, respectively, according to limb and GT morphology). We found that both *Fgf10* and *Fgfr2* expressions are mainly restricted to the urethral plate (Figure 6F–L). Interestingly, *Fgf10* expression was also found in the dorsal and ventral regions of labioscrotal swellings in E26 guinea pig GT and was confined to four small, round areas (Figure 6H,I). The *Fgf10* expression in labioscrotal swellings has never been reported before, and the function of *Fgf10* in these domains is unidentified.

The *Wnt5a* expression domain is located in the mesenchyme of the GT, with the strongest expression at the distal tip. From both ventral and dorsal views, we can clearly see that *Wnt5a* expression is mainly located in the distal glans region (Figure 5Q,R), suggesting it may control the outgrowth of the glans at an early stage, as reported in mouse GT [21,36].

Taken together, these results reveal dynamic patterns of key developmental gene expression in guinea pig external genitalia, with *Fgf8*, *Fgfr2*, *Bmp4*, *Bmp7*, *Hoxd13,* and *Wnt5a* showing similar expression patterns during GT development between guinea pigs and mice. We identified that *Shh* and *Fgf10* show differential expression patterns between the two species, which may contribute to the differential morphogenesis of external genitalia between guinea pigs and mice.

### 3.4. Relative Expression Levels of Shh, Fgf10, and Fgfr2 in Developing Guinea Pig GT Are Reduced Compared with the Comparable Stage of Mouse GT

Compared with mice, the major difference in guinea pig GT development is that the urethral plate opens to form the urethral groove before sexual differentiation (Figure 1, Figure 2 and Figure 3, Refs. [3,5]). Shh, Fgfs, Hoxd13, Hoxa13, Bmp4, and Wnt pathways play key roles in glans and preputial morphogenesis [11,12,13]. In light of our findings regarding the differential expression of *Hoxd13* and *Fgf8* between GT and limb buds and between guinea pigs and mice, we hypothesized that the expression levels of these developmental genes in developing guinea pig GT may be relatively weaker than those in mouse GT. To verify this hypothesis, we tested and compared the relative expression levels of these genes between mouse and guinea pig GTs with comparable developmental stages. Developing limb buds were widely used to determine the developmental stages in different species [37,38]. According to limb (Figure 7A–D) and external genital (Figure 7E–H) morphology, we found that E23 and E26.5 GTs of guinea pigs are comparable to E12.5 and E13.5 GTs of mice, respectively. As we compare relative gene expression levels between two different species, the selection of housekeeping genes is important. We compared Ct values of commonly used housekeeping genes, *Gapdh* and *Actb,* and found that the Ct values of *Gapdh* were closer to each other between mouse and guinea pig GTs and more stable when the same amount of total RNA was applied (Appendix A). Thus, *Gapdh* was selected as the housekeeping gene to quantify the gene expression levels. Compared to the two housekeeping genes, the Ct values for all the selected developmental genes—*Shh*, *Fgf8*, *Fgf10*, *Fgfr2*, *Hoxd13*, *Hoxa13*, *Bmp4,* and *Ctnnb1*—at all the tested stages were found to be higher. Compared with E12.5 and E13.5 mice, lower expression levels of all eight developmental genes were detected in the GTs of E23 and E26.5 guinea pigs (*p* ≤ 0.0061, *n* = 5). *Hoxa13*, *Bmp4*, and *Ctnnb1* were about 2-fold lower in guinea pig GTs compared with mice, and the expression of *Shh*, *Fgf8, Fgf10, Fgfr2,* and *Hoxd13* was more than 4.5-fold lower compared with mice (Figure 7I,J). We also found that *Actb* relative expression in developing GTs showed no significant difference (E23, 1.23-fold, *p* = 0.163, *n* = 5; E26.5, 1.14-fold, *p* = 0.248, *n* = 5) between guinea pigs and mice (Figure 7I,J). Compared with the expression in E12.5 mice, *Shh*, *Fgf8, Fgf10, Fgfr2,* and *Hoxd13* in E23 guinea pig GTs were downregulated by 5.54-, 4.5-, 14.62-, 6.47-, and 7.51-fold, respectively (Figure 7I). Likewise, all the five genes were dramatically upregulated in E13.5 mouse GT (*Shh*, 6.23-fold; *Fgf8*, 5.9-fold; *Fgf10*, 9.65-fold; *Fgfr2*, 7.24-fold; *Hoxd13*, 9.98-fold) compared with E26.5 guinea pigs (Figure 7J). Because these are relative gene expression levels in two species, we believe that the approximately 2-fold reductions in *Hoxa13* (E23, 1.9-fold; E26.5, 2.8-fold), *Bmp4* (E23, 2.1-fold; E26.5, 2.6-fold), and *Ctnnb1* (E23, 2.5-fold; E26.5, 2.4-fold) may result from systemic differences. The dramatic differences (more than 4.5-fold) in the expression of *Shh*, *Fgf8, Fgf10, Fgfr2,* and *Hoxd13* may play roles in patterning cellular processes and lead to differential morphological developments such as urethral groove formation and preputial development between the two species.

### 3.5. Shh and Fgf10/Fgfr2 Play Key Roles in Preputial and Urethral Groove Formation

Shh is a chief morphogen that organizes the structure in the ventral midline of multiple organs [6,39,40], and the deletion of Shh signaling genes at E13.5–15.5 led to an open urethral plate (hypospadias) and a reduction in glans and preputial development in mice [33,41]. Fgf10 and its receptor Fgfr2 also play important roles in urethral and preputial development [15,16,42]. Additionally, Shh and Fgf10/Fgfr2 signaling interact with the expression of other developmental genes, such as *Fgf8*, *Hoxd13*, *Bmp4,* and *Wnt5a*, to coordinately regulate morphogenesis during GT development; e.g., *Shh* negatively regulates Bmp and Wnt signaling molecules, which will define the limits of the range of each domain [11,40]. Based on our findings that *Shh* expression levels were reduced more than 5-fold in the developing GT of guinea pigs compared to that of mice, and considering the difference in *Shh* expression patterns between mouse and guinea pig GTs (Figure 6A–E and Ref. [5]), we observed that the *Fgf10* expression pattern shifts from the mesenchyme of mouse GT to mainly the urethral epithelium of guinea pig GT. Furthermore, expression levels in guinea pig GT reduce more than 14-fold at E23 (compared with E12.5 mouse GT) and more than 9-fold at E26.5 (compared with E13.5 mouse GT). We hypothesized that the reduction in Shh and Fgf10/Fgfr2 signaling may induce urethral groove formation and delay preputial development in guinea pigs. To test this hypothesis, we performed GT organ culture and examined the roles of *Shh* and *Fgf-10* in urethral groove and preputial formation using Fgf-10 and Shh proteins, the Fgf receptor inhibitor NVP-BGJ398, and the hedgehog signal inhibitor cyclopamine. Male GTs of E14.5 mice and E27 guinea pigs were dissected and maintained in culture using media supplemented with or without androgen (10 nM MT), Shh and Fgf10/Fgfr2 inhibitors (for mouse GT culture), and Shh and Fgf-10 proteins (for guinea pig GT culture) to test the function of Shh and Fgf-10 in urethral groove and preputial development. We first observed the effect on androgen’s mouse GT culture. After 48 h of culture, both the control male mouse GTs with and without androgen developed preputial swellings and a urethral plate (Figure 8A–C). However, compared to the GTs with 10 nM MT, there was a small hole remaining open in the middle of the urethral plate on the ventral side of the GTs without MT (Figure 8B). Thus, we added 10 nM MT in our male GT culture system when testing the effects of Shh, Fgf10/Fgfr2, and the inhibitors. We then performed the same stage of male mouse GT culture with the administration of 1.4 nM BGJ398, 200 nM cyclopamine, or 1.4 nM BGJ398 plus 200 nM cyclopamine to the media from the beginning. All three groups of GTs formed smaller preputial swellings and widely opened urethral grooves compared with controls (Figure 8D,E). After 48 h of culture with both 1.4 nM BGJ398 and 200 nM cyclopamine, the preputial swelling almost disappeared, and the GT grew its shape into a cylinder with an open urethral groove; in addition, the distal tip also malformed (Figure 8F). Next, we performed guinea pig GT culture (starting from E27) with or without 10 nM MT (Figure 8G–I). Because the guinea pig GT is very soft and larger/heavier compared with the mouse GT at this stage, when being cultured in a ventral side-up position, the distal part of the GT became flat, making it difficult to overcome the gravity to form a closed urethra using the same method as for mouse GT culture. We believe that additional support material should be applied in our guinea pig GT culture system in the future. After the administration of 50 ng/mL Fgf-10 protein, the guinea pig GT enlarged in size compared with the control and formed preputial swellings after 48 h, which have never been seen in controls (Figure 8J). After 48 h of culture with 200 ng/mL Shh protein, the guinea pig GT also formed preputial swellings; however, the elongation of the GT was not as obvious as that of the Fgf-10-treated ones (Figure 8K). When male guinea pig GT was cultured with both 50 ng/mL Fgf-10 and 200 ng/mL Shh proteins, the preputial swellings became more obvious after 48 h (Figure 8L). Our results suggest that Shh and Fgf10/Fgfr2 signaling play key roles in the urethral groove and preputial formation during penile development.

## 4. Discussion

During human penile development, the process of tubular urethral formation has been described as distal-opening-proximal-closing [1]. To date, the guinea pig is the only published animal model of penile development in the literature that can demonstrate a process similar to what is seen in humans [3]. The most commonly used animals, mice and rats, form urethral plates and a urethral opening at the ventral proximal region, rather than a fully open urethral groove like humans and guinea pigs [4,6]. From our perspective, preputial development in guinea pigs is more similar to the human preputial development model proposed by Glenister [8], and the detailed process of distal extension of the prepuce in guinea pigs is similar to a modified human preputial development model described by Cunha [9], suggesting that preputial development in guinea pigs and humans may share similar mechanisms. Interestingly, Liu et al. [43] reported that mouse external and internal prepuces occur via entirely different morphogenetic mechanisms. The structure they called internal prepuce develops postnatally, and humans have no similar structure in penile development. Based on our observations, the external prepuce development in E16.5 mice shows a similar developmental process to that of guinea pigs, with increasing epithelial layers and preputial mesenchyme evaginating distally. The major difference in preputial development between mice and guinea pigs is the timing of preputial development relative to tubular urethra formation. In guinea pigs, and humans as well, the urethral groove forms in the glans penis without preputial covering; in fact, the fully opened urethral groove forms at E28 in guinea pigs [3,5] and at 9.5 weeks of gestation in humans [1], before the initiation of preputial development in the proximal region of the glans penis (Figure 2 and Ref. [9]). In mice, the opened urethral groove without preputial covering can only be observed in the distal region of the E16.5 mouse penis (Figure 2 and Ref. [43]), suggesting a delay in the relative timing of preputial development to the formation of an opened urethral groove in guinea pigs and humans compared with mice.

Genetic control of external genital development has been well studied in mice, and genes in Shh, Fgf, Bmp, and Wnt pathways were found to be expressed in developing mouse GT before sexual differentiation. We compared these pathway genes in developing GT between guinea pigs and mice and found that the mRNA expression patterns of *Shh* and *Fgf10* were different, while the expression levels of *Shh*, *Fgf8*, *Fgf10*, *Hoxd13*, *Bmp4,* and an important canonical Wnt signaling gene, *Ctnnb1*, were reduced in guinea pigs compared with stage-matched mouse GT. Shh signals are proposed to the ventral ectoderm to maintain the structural integrity of the epithelium, which is essential for the maintenance of a closed urethral tube [11]. Preputial *Shh* expressions were found almost at the same time of epithelial folding to form the preputial lamina in mice [6], and deletion of the Shh pathway gene *Smo* using Msx2cre disrupted urethral plate and preputial development [41]. If we compare normal human [44] and guinea pig [3] GTs with those of *Smo* conditional knockout mutant mice [41], they all have an open urethral groove, and the mutant mouse GT has a malformed prepuce (Figure 9A–D). The shift in the *Shh* expression domain from the urethral epithelium to the ventral ectodermal epithelium in developing guinea pig GT [5] disrupts the signals to ventral ectoderm, which may be one of the reasons that lead to urethral opening and urethral groove formation. Preputial *Shh* mRNA expression (Figure 7) was first found (around E31) two days after the initiation of preputial development (E28-E29), suggesting that *Shh* may play a role in preputial development, but not in its initiation, in guinea pigs.

*Fgf8*, *Fgf10,* and their receptor *Fgfr2* were identified as candidate genes for human hypospadias [45,46]. In mice, several Fgf ligands and their receptors, including Fgf8, Fgf10, and Fgfr2 are expressed in multiple domains in developing GT [47]. *Fgf8* is more likely a readout and not required for the outgrowth or normal patterning of the GT in mice [35], but *Fgf10* and *Fgfr2* have been revealed to play key roles in urethral tube closure and normal preputial development [15,16,42]. In mice, mesenchymal Fgf10 interacts with urethral epithelial Fgfr2 (Fgfr2IIIb) to maintain normal GT development with an unopened urethral plate [42,48]. Deletion of *Fgf10* or *Fgfr2IIIb* leads to severe hypospadias in mice, in which the ventral side of the urethra is fully open, resembling a urethral groove in guinea pigs, but still with the malformed prepuce [15,42,48]. Loss-of-function mutations or deletions of *Fgfr2* in the ectoderm result in the most severe hypospadias; the *Fgfr2* mutant mice exhibit an open urethra [3,16,48], and the morphology of the developing GT looks similar to that of guinea pig [3] and human [44] GTs at the proximal-open stage, while the distal remains closed (Figure 9A,B,E). Our data showed differential expression patterns and levels of *Shh*, *Fgf10,* and *Fgfr2* in GT between guinea pigs and mice, suggesting that *Shh* and *Fgf10/Fgfr2* play key roles in urethral groove formation in guinea pigs, and possibly in humans as well. If we compare the morphology of normal human and guinea pig developing GTs with the GTs of *Shh* and *Fgfr2* conditional knockout mutant mice, we can see the similarities; both mutant GTs showed an open urethral groove and a reduction in prepuce (Figure 9).

*Hoxd13* is the gene with the most reduced expression level in the GT of guinea pigs compared with that of mice (Figure 7I,J). According to Lin et al. [41], *Hoxd13* has been reported to be one of the downstream genes of *Shh* in the external genital and limb development [41,49]. Shh and Fgf signaling were found working together to control the expression of *Hoxd* genes in limb development [50]. Thus, we believe the reduction in the *Hoxd13* expression in guinea pigs developing GT may be induced by the decreased Shh, Fgf8, Fgf10, and Fgfr2 signaling.

Our organ culture results clearly showed that both Shh and Fgf10/Fgfr2 signaling are required to maintain the prepuce and urethral plate in the developing mouse GT. The lack of either one will induce an open urethral groove (Figure 8). Block Shh or Fgf signaling has a detrimental effect on preputial development, and blocking both pathways results in the most severe reduction in developing prepuce (Figure 8). In the guinea pig GT culture system, since the E27 GT is much larger and softer compared with the E14.5 mouse GT, gravity caused the GT to grow into a kind of flat shape after 2 days of culture. We found that all the GTs (100%) formed preputial swellings in our Shh and Fgf-10 protein-treated groups, although they all have a fully opened urethral groove. Considering the delayed preputial *Shh* expression (Figure 6D,E) and the delayed preputial development in guinea pigs (Figure 2), we predict that, compared with mice, the ectodermal Fgf signaling may also be expressed later, coinciding with the initiation of proximal urethral closure, in response to the androgen signaling. During the development of the human penis, tubular urethral formation follows a process of distal-opening-proximal-closing, with the formation of the urethral groove being the first step. We hope that our findings can provide a basis for further research on the mechanism of distal-opening-proximal-closing, and provide a reference for clinical treatment of penile abnormalities such as hypospadias.

## Figures and Tables

**Figure 1 cells-14-00348-f001:**
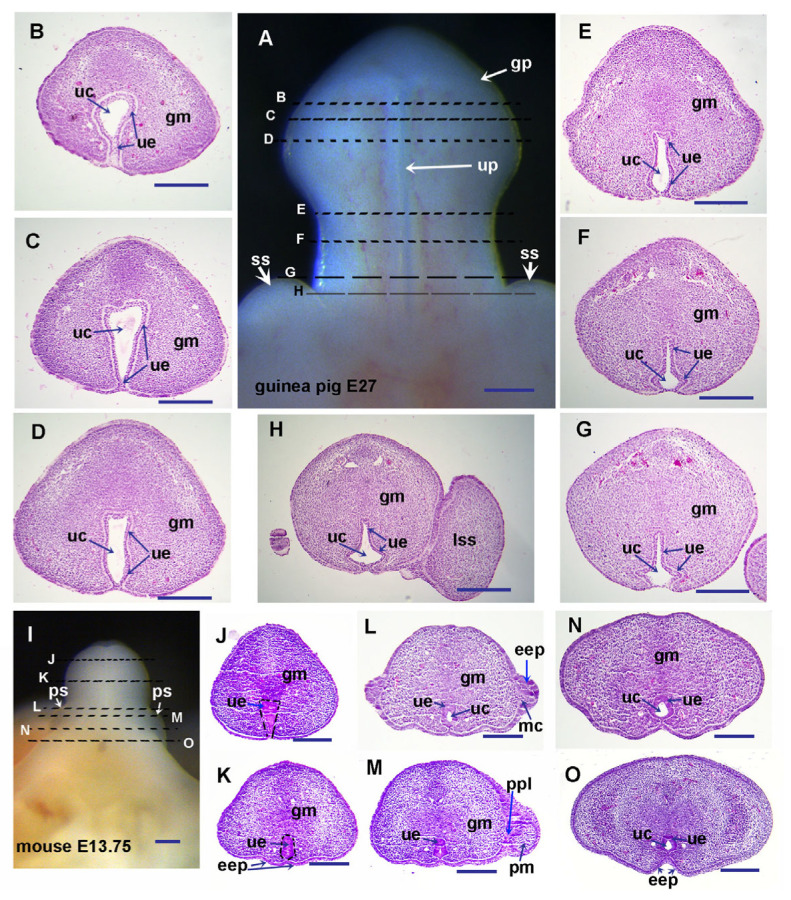
Histological structure of E27 guinea pig and E13.75 mouse genital tubercles (GTs). Images (**A**,**I**) are ventral views of E27 guinea pig (**A**) and E13.75 mouse (**I**) GTs with distal at the top. All sections of guinea pig (**B**–**H**) and mouse (**J**–**O**) are transverse through GT with dorsal at the top. Broken lines on images (**A**,**I**) indicate the planes of sections. Note the prepuce starts to form in E13.75 mice, but not in E27 guinea pigs. Abbrev: eep, epidermal epithelium; gm, glans mesenchyme; gp, glans penis; mc, mesenchyme; pm, preputial mesenchyme; ppl, preputial lamina; ps, preputial swelling; ss, scrotal swelling; uc, urethral canal; ue, urethral epithelium; up, urethral plate. Scale bars: 250 µm.

**Figure 2 cells-14-00348-f002:**
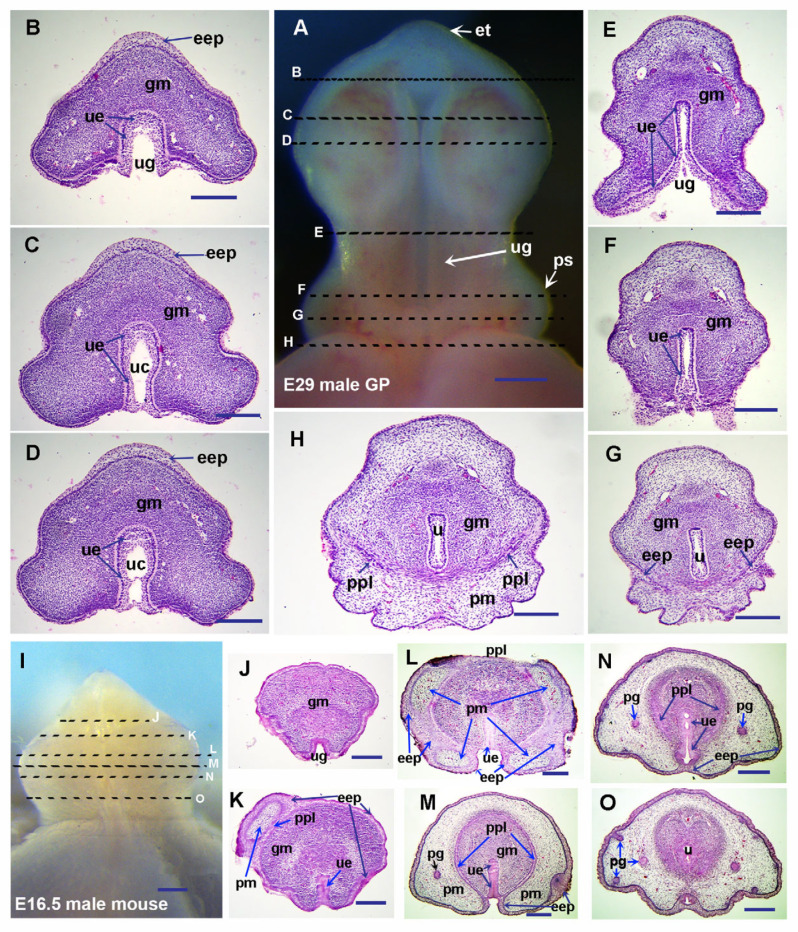
Histological structure of E29 guinea pig (GP) and E16.5 mouse penises. Images (**A**,**I**) are ventral views of E29 guinea pig (**A**) and E16.5 mouse (**I**) penises, with distal at the top. All sections of the guinea pig (**B**–**H**) and mouse (**J**–**O**) are transverse through penises, with dorsal at the top. Broken lines on images (**A**,**I**) indicate the planes of sections. Note that the prepuce starts to form at E29 in guinea pigs but reaches to the distal glans penis (**K**) at E16.5 in mice. Abbrev: et, epithelium tag; pg, preputial gland; u, urethra; ug, urethral groove; eep, gm, pm, ppl, ps, ss, uc, and ue are the same as in Figure 1. Scale bars: 250 µm.

**Figure 3 cells-14-00348-f003:**
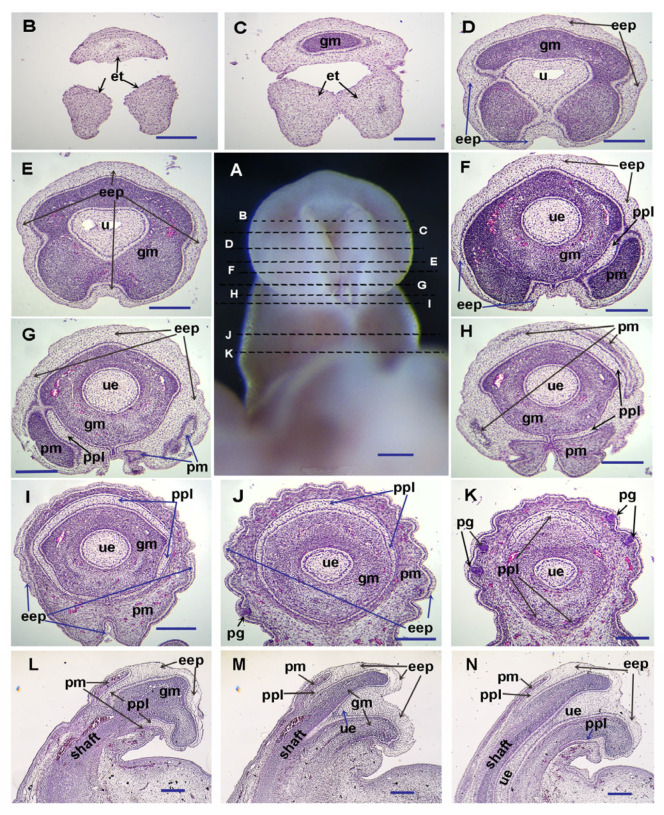
Histological structure of the E33 guinea pig penis. Image (**A**) shows a ventral view of the E33 guinea pig penis with the distal end at the top. Sections of (**B**–**K**) are transverse through the penis, with dorsal at the top. Broken lines on the image (**A**) indicate the planes of section. Sections of (**L**–**N**) show sagittal planes of E33 guinea pig penis with dorsal at the left. Note that the prepuce reaches the distal glans penis at E33 in guinea pigs. Abbrev: eep, gm, pm, ppl, and ue are the same as in Figure 1; et, pg, and u are the same as in Figure 2. Scale bars: 250 µm.

**Figure 4 cells-14-00348-f004:**
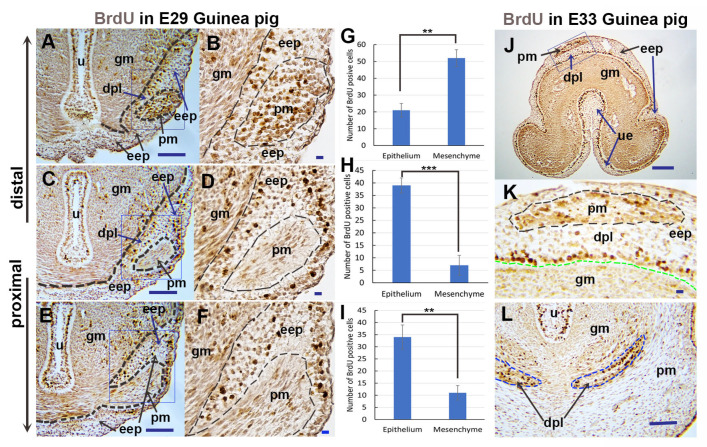
Cell proliferation in guinea pig preputial development. Images show immunolocalization of BrdU (dark brown) on transverse sections of guinea pig penises at E29 (**A**–**F**) and E33 (**J**–**L**) with dorsal at the top. (**A**,**C**,**E**) are sections through the proximal region (See image in Figure 2A, between plane F and H) of the E29 guinea pig penis with the order from proximal to more distal as (**E**), (**C**), and (**A**). (**B**,**D**,**F**) are higher magnification images of the areas marked by blue boxes in (**A**,**C**,**E**), respectively. (**G**,**H**,**I**) show the number of BrdU-positive cells in a newly formed small region of preputial mesenchyme (pm) and surrounding epidermal epithelium (eep) counted from (**B**,**D**,**F**), respectively. Note that epidermal epithelia cells proliferate and invaginate to form original preputial lamina (**B**,**F**,**I**) and separate a small portion of preputial mesenchyme (**C**,**D**,**H**), and then the preputial mesenchymal cells proliferate and evaginate distally (**A**,**B**,**G**). (**J**,**L**) are distal (**J**) and proximal (**L**) sections of the E33 guinea pig penis, and (**K**) is a higher magnification image of the area marked by a blue box in (**J**). Note that the basal layer of epithelial cells (the deepest layer above the green line) proliferates to increase epidermal epithelial cell layers, and preputial mesenchymal cells (inside the black line) proliferate to evaginate distally (**J**,**K**), In the proximal section (**L**), BrdU-positive cells can be observed in developing preputial lamina (inside blue line). The data in (**G**–**I**) are mean (*n* = 3) ± standard error (SE), ** *p* < 0.01, *** *p* < 0.001. Abbrev: dpl, developing preputial lamina; eep, gm, pm, and ue are the same as in Figure 1, u is the same as Figure 2. Scale bars: 100 μm.

**Figure 5 cells-14-00348-f005:**
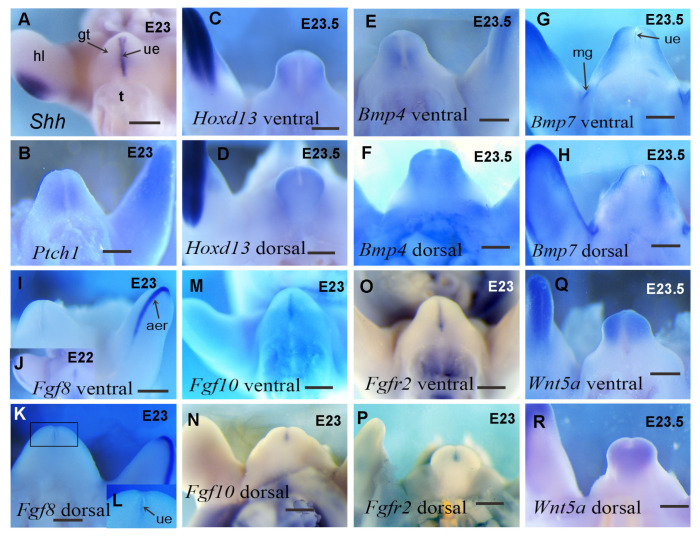
In situ gene expression analysis of guinea pig genital tubercle (GT). Images are ventral (**A**–**C**,**E**,**G**,**I**,**J**,**M**,**O**,**Q**) or dorsal (**D**,**F**,**H**,**K**,**L**,**N**,**P**,**R**) views of E23-E23.5 guinea pig (excerpt for J, which is E22) GTs with the distal at the top. The tail has been removed from all embryos. Purple or blue staining in each image indicates the gene expression domain. Developmental stages are labeled upright in each image. Note that *Shh* is expressed in the urethral epithelium (**A**) and its receptor is expressed in the adjacent mesenchyme (**B**). *Hoxd13* (**C**,**D**), *Bmp4* (**E**,**F**), and *Wnt5a* (**Q**,**R**) are expressed in genital mesenchyme. *Bmp7* is expressed in the distal urethral epithelium (very weak), genital mesenchyme, and developing mammary glands (**G**,**H**). *Fgf8* expression is located in the urethral epithelium at E22 (**J**), but it is only weakly expressed in the distal tip part of the urethral epithelium and can only be seen in dorsal view at E23 (**I**,**K**,**L**). *Fgf10* expression in genital mesenchyme is very weak and is relatively strong in urethral epithelium (**M**,**N**). *Fgfr2* expression is mainly in the urethral epithelium (**O**,**P**). Abbrev: aer, apical ectodermal ridge; gt, genital tubercle; hl, hindlimb; mg, mammary gland; ue is the same as Figure 1. Scale bars: 500 µm.

**Figure 6 cells-14-00348-f006:**
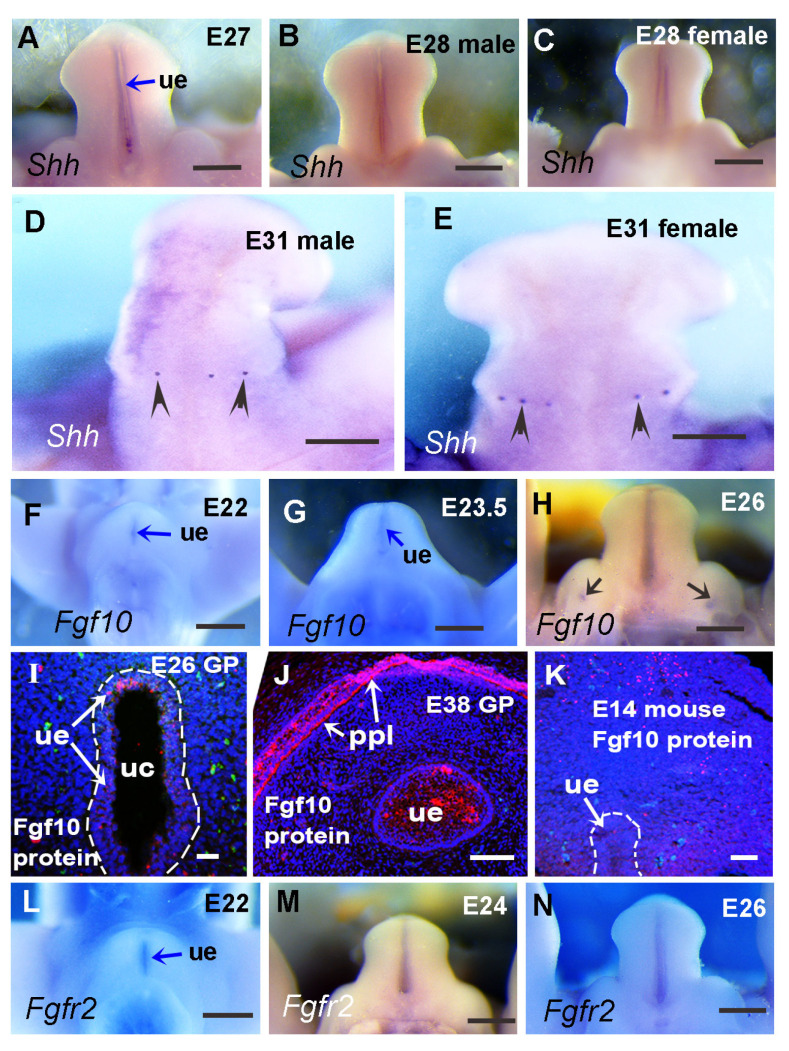
Spatiotemporal patterns of *Shh*, *Fgf10*, and *Fgfr2* expression during guinea pig external genital development. Whole-mount images are ventral (**A**–**C**,**F**–**H**,**L**–**N**) or dorsal (**D**,**E**) views of the developing external genitalia of guinea pigs (**E**,**G**) with the distal at the top, showing mRNA expression of *Shh* (**A**–**E**), *Fgf10* (**F**–**H**) and *Fgfr2* (**L**–**N**). Images in (**I**–**K**) are transverse sections of the developing penises of guinea pigs and mice, showing Fgf10 protein (in red; blue is Dapi) localization. Note that *Shh* mRNA is exclusively expressed in the urethral epithelium and prepuce during the later stages (**A**–**E**) of developing EG in guinea pigs. *Fgf10* mRNA and protein mainly localize in the urethral epithelium at early stages in the guinea pig genital tubercle (GT) (**F**–**I**) and also in the preputial lamina (ppl) in E38 penis (**J**) of guinea pigs. However, in mouse GT, Fgf10 protein mainly localizes in the mesenchyme. *Fgfr2* mRNA is mainly expressed in the urethral epithelium in guinea pig GT. Arrowheads in (**D**,**E**) point to the tiny *Shh* expression domain, while arrows in (**H**) indicate the *Fgf10* expression domain in labioscrotal swellings. Abbrev: ppl and ue are the same as in Figure 1. Scale bars in (**A**–**H**,**L**–**N**): 500 µm; in (**I**–**K**): 100 µm.

**Figure 7 cells-14-00348-f007:**
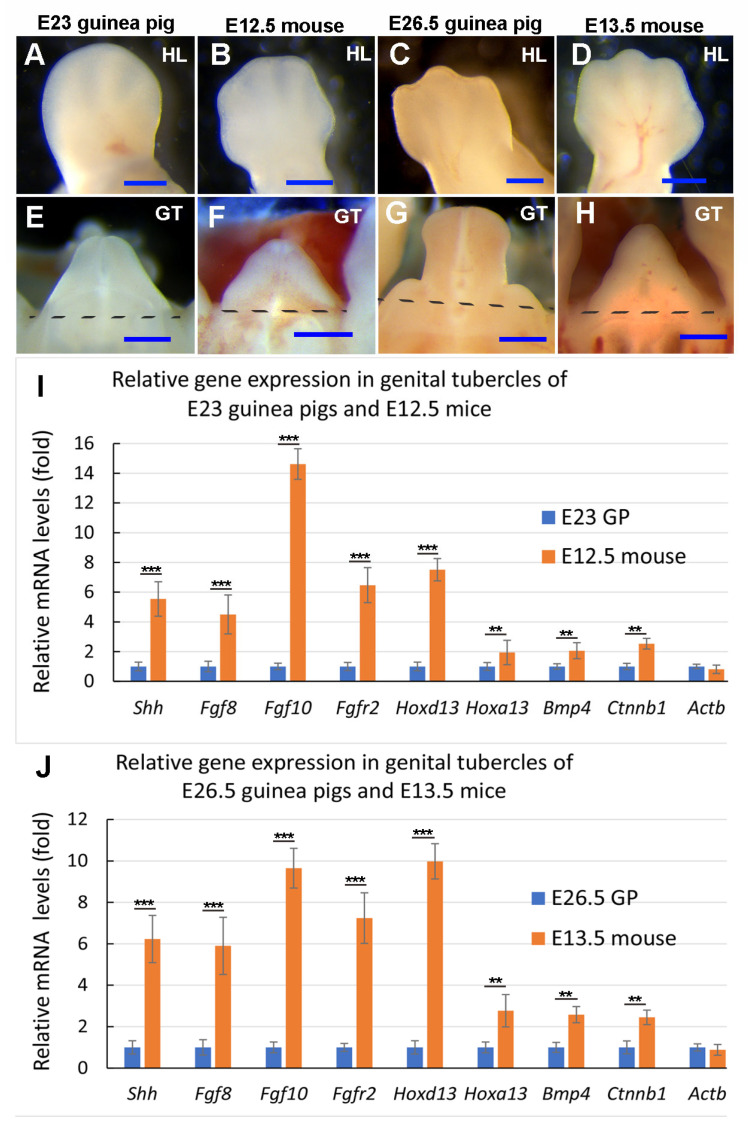
Comparison of relative expression levels of key developmental genes between guinea pig (GP) and mouse genital tubercles (GTs). Images in (**A**–**H**) are GP and mouse hindlimb (**H**) buds (**A**–**D**) and GTs (**E**–**H**), showing developmental stage similarity. All limb buds are in a dorsal view with the anterior on the left, and all GTs are in a ventral view with the distal at the top. GT samples were cut from the level marked by dashed lines in (**E**–**H**) for RNA extraction and quantitative PCR analysis. Data in (**I**,**J**) are mean ± standard error, *n* = 5, ** *p* ≤ 0.01, *** *p* ≤ 0.001. Scale bars in (**A**–**H**): 500 µm.

**Figure 8 cells-14-00348-f008:**
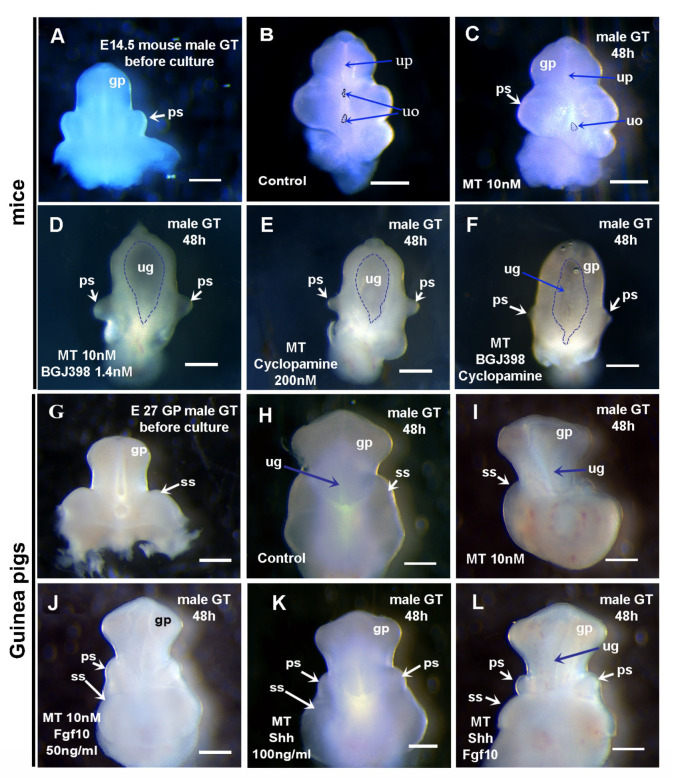
The effect of Shh and Fgf-10 proteins on urethral groove and preputial development. Images in (**A**–**L**) are ventral views of mouse (**A**–**F**) and guinea pig (**G**–**L**) genital tubercles (GTs) with the distal end at the top. Broken line in images (**D**–**F**) shows the edge of urethral groove. Note that Hedgehog or Fgf inhibitors induce urethral groove formation but inhibit preputial development in cultured GTs of mice; the most significant effect is seen in cultured GTs with both inhibitors (**D**–**F**). Shh or Fgf-10 protein induces the formation of preputial swellings in cultured GTs of guinea pigs, and the most obvious results were found in both protein-treated GTs (**J**–**L**). Abbrev: uo, urethral opening; gp, ps, ss, and up are the same as in Figure 1, ug is the same as in Figure 2. Scale bars: 500 µm.

**Figure 9 cells-14-00348-f009:**
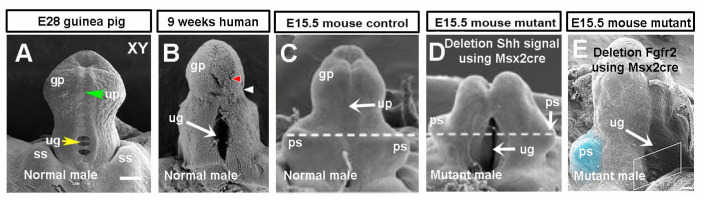
Conditional deletion of *Smo* and *Fgfr2* in mouse genital tubercle (GT) resulted in a fully opened urethral groove resembling a normal human and guinea pig developing penis. Images (**A**–**E**) are modified from published figures with permission. (**A**) Guinea pig E28 GT [5]. (**B**) Human 9-week-old developing penis [44]. (**C**,**D**) Mouse E15.5 normal male GT (**C**) and Msx2-rtTA;tetO-Cre;Smoc/c male GT (**D**) [41]. (**E**) Mouse E15.5 Msx2cre Fgfr2 c/c male GT [16]. Abbrev: gp, ps, ss and up are the same as in Figure 1, ug is the same as in Figure 2.

## Data Availability

All data are available upon reasonable request to the corresponding author.

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
