# Peer review of "Differences in Formation of Prepuce and Urethral Groove During Penile Development Between Guinea Pigs and Mice Are Controlled by Differential Expression of Shh, Fgf10 and Fgfr2"

_cells, 2025, doi:10.3390/cells14050348_

Round 1
Reviewer 1 Report
Comments and Suggestions for Authors
An interesting, richly illustrated work. Interesting results of the development of the urinary system are presented in relation to the pathologies that accompany it. The results are available during research on the urinary system in connection with treatment in humans, because the process of development in the studied animals has been described and documented. This manuscript deserves on publication.
Author Response
An interesting, richly illustrated work. Interesting results of the development of the urinary system are presented in relation to the pathologies that accompany it. The results are available during research on the urinary system in connection with treatment in humans, because the process of development in the studied animals has been described and documented. This manuscript deserves on publication.
Response:
We thank the reviewer for his/her positive comments on our manuscript.
Reviewer 2 Report
Comments and Suggestions for Authors
The manuscript is about evaluation of differential expression of Shh, Fgf10 and Fgfr2 related to different formacion of urethral groove and prepuce in guinea pigs and mice. It is a basic research article. I found the manuscript very interesting and well written. And add info about development of these structures in animal models
Even methods are well designed and clear
Discussion and conclusions are well drawn and balanced.
No ethics concerns
I would implement the limitations of the study and better highlight the possible impact of your research on future.
Author Response
The manuscript is about evaluation of differential expression of Shh, Fgf10 and Fgfr2 related to different formation of urethral groove and prepuce in guinea pigs and mice. It is a basic research article. I found the manuscript very interesting and well written. And add info about development of these structures in animal models
Even methods are well designed and clear
Discussion and conclusions are well drawn and balanced.
No ethics concerns
I would implement the limitations of the study and better highlight the possible impact of your research on future.
Response: We thank the reviewer for his/her positive comments. About the limitations in guinea pig GT organ culture, we are working on finding a suitable support material for the future research. Our next step will be focused on the mechanism of distal-opening-proximal-closing.
Reviewer 3 Report
Comments and Suggestions for Authors
We have carefully read the manuscript cells-3432277 entitled ‘Differences in formation of prepuce and urethral groove during penile devel-opment between guinea pigs and mice are controlled by differential expression of Shh, Fgf10 and Fgfr2’ by Dr Wang and Zheng from the Southern Illinois University.
From the observation that the humans and guinea pigs share similar mechanisms of penile devlopment contrarily to mice, the authors study the pattern of gene expression in these two animal species. They found that several genes are expressed diffrently in guinea pigs and mice, especially, Shh, Fgf10 and Fgfr2 and this differential expression can explain, at least in part, the mechanism of the formation of the urethral groove during the penile development.
The manuscript is well written and easy to follow with a lot of data in support of the conclusions formulated by the authors. The introduction is complete and very informative to allow the readers to understand the results and then the conclusions.
I have just one comment about the manuscript. The authors could better describe the potential applications of their findings in relation to guidance for cell differentiation in iPS-derived culture to reconstruct organs (a probably long-term perspective of tissue engineering), the potential impact on pathologies such as the effect of endocrine disruptor chemicals on penile devlopment.
Nevertheless it remains an excellent manuscript supported y strong evidences.
Author Response
We have carefully read the manuscript cells-3432277 entitled ‘Differences in formation of prepuce and urethral groove during penile development between guinea pigs and mice are controlled by differential expression of Shh, Fgf10 and Fgfr2’ by Dr Wang and Zheng from the Southern Illinois University.
From the observation that the humans and guinea pigs share similar mechanisms of penile development contrarily to mice, the authors study the pattern of gene expression in these two-animal species. They found that several genes are expressed differently in guinea pigs and mice, especially, Shh, Fgf10 and Fgfr2 and this differential expression can explain, at least in part, the mechanism of the formation of the urethral groove during the penile development.The manuscript is well written and easy to follow with a lot of data in support of the conclusions formulated by the authors. The introduction is complete and very informative to allow the readers to understand the results and then the conclusions.
I have just one comment about the manuscript. The authors could better describe the potential applications of their findings in relation to guidance for cell differentiation in iPS-derived culture to reconstruct organs (a probably long-term perspective of tissue engineering), the potential impact on pathologies such as the effect of endocrine disruptor chemicals on penile development.Nevertheless, it remains an excellent manuscript supported by strong evidences.
Response: We thank the reviewer for his/her careful reading of the manuscript and the constructive comments, and we agree that the potential impact should be addressed. We then added the following 2 sentences at the end of the article. “During the development of human penis, tubular urethral formation follows a process of distal-opening-proximal-closing, and the formation of urethral groove is the first step. We hope that our findings can provide a basis for further research on the mechanism of distal-opening-proximal-closing and provide a reference for clinical treatment of penile abnormalities such as hypospadias.”